# TextCtrl: Diffusion-based Scene Text Editing with Prior Guidance Control

**Weichao Zeng**[1,3]  **Yan Shu**[1]  **Zhenhang Li**[1,3]
**Dongbao Yang**[1,3]  **Yu Zhou**[2*]

[1] Institute of Information Engineering, Chinese Academy of Sciences
[2] VCIP & TMCC & DISSec, College of Computer Science, Nankai University
[3] School of Cyber Security, University of Chinese Academy of Sciences
{zengweichao, lizhenhang, yangdongbao}@iie.ac.cn
shuyan9812@gamil.com, yzhou@nankai.edu.cn

## Abstract

Centred on content modification and style preservation, Scene Text Editing (STE) remains a challenging task despite considerable progress in text-to-image synthesis and text-driven image manipulation recently. GAN-based STE methods generally encounter a common issue of model generalization, while Diffusion-based STE methods suffer from undesired style deviations. To address these problems, we propose **TextCtrl**, a diffusion-based method that edits text with prior guidance control. Our method consists of two key components: (i) By constructing fine-grained text style disentanglement and robust text glyph structure representation, TextCtrl explicitly incorporates Style-Structure guidance into model design and network training, significantly improving text style consistency and rendering accuracy. (ii) To further leverage the style prior, a Glyph-adaptive Mutual Self-attention mechanism is proposed which deconstructs the implicit fine-grained features of the source image to enhance style consistency and vision quality during inference. Furthermore, to fill the vacancy of the real-world STE evaluation benchmark, we create the first real-world image-pair dataset termed **ScenePair** for fair comparisons. Experiments demonstrate the effectiveness of TextCtrl compared with previous methods concerning both style fidelity and text accuracy. Project page: https://github.com/weichaozeng/TextCtrl.

## 1 Introduction

Scene Text Editing (STE) refers to modifying the text with desired content on an input image while preserving the styles and textures of both the text and the background to maintain a realistic appearance [1]. As a newly emerging task in the field of scene text processing [2], STE not only possesses distinctive application value [3, 4, 5] but also benefits the text-oriented downstream research in detection [6], recognition [7, 8], spotting [9] and reasoning [10, 11]. Recently, increasing attention has been paid to GAN-based and diffusion-based scene text editing methods.

Exploiting the Generative Adversarial Networks (GANs) [12], early works [1, 13] decompose STE into three subtasks: foreground text style transfer, background restoration and fusion. The divide-and-conquer manner significantly reduces the difficulty of pattern learning and enables the pre-training of sub-modules with additional supervision [14]. However, the generalization capabilities of these methods are inevitably limited due to the constrained model capacity of GANs [15] and the challenges in accurately decomposing text styles [4]. Besides, as observed in experiments, the divide-

---

*Corresponding Author

38th Conference on Neural Information Processing Systems (NeurIPS 2024).

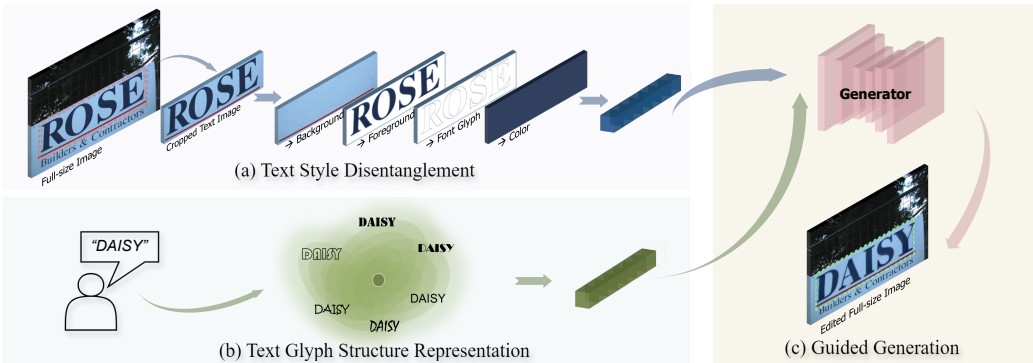

Figure 1: Conceptual illustration of the decomposition of STE by TEXTCTRL. (a) Text style is disentangled into text background, text foreground, text font glyph and text color features. (b) Text glyph structure is represented by the cluster centroid of various font text features. (c) The explicit style features and structure features guide the generator to perform scene text editing.

and-conquer design brings about the "bucket effect", wherein the unstable background restoration quality leads to messy fusion artifacts.

Recently, large-scale text-to-image diffusion models [16, 17] have convincingly demonstrated strong capabilities in image synthesis and processing. Several methods attempt to realize STE in a conditional synthesis manner, including style image concatenation [18] and one-shot style adaptation [19]. However, these methods are limited to **the coarse-grained learning of miscellaneous styles** from text images. Other methods [20, 21, 22] tend to resolve STE in a universal framework along with STG (Scene Text Generation) in an inpainting manner conditioned on full images, which enables the leverage of large-scale data for self-supervised learning [23]. Nevertheless, their **style guidance predominantly originates from the image's unmasked regions**, which can be unreliable in complex scenarios and fail in style consistency. Besides, resulting from **the weak correlation between text prompt and glyph structure** [24, 25, 26], diffusion-based STE methods are prone to generating typos, which decreases the text rendering accuracy.

For the aforementioned problems, we identify insufficient prior guidance on both style and structure as the primary factor that impedes the previous methods from performing accurate and faithful scene text editing. As depicted in Fig. 1, we propose a conditional diffusion-based STE model, wherein our method decomposes the prerequisite of STE into two main aspects: text style disentanglement and text glyph representation. The fine-grained disentangled text style features ensure visual coherency, while the robust glyph structure representation improves text rendering accuracy. The dual Style-Structure guidance collectively contributes to significant enhancements in STE performance.

Furthermore, undesired color deviation and texture degradation compared with the source text image occasionally occur in the inference of diffusion-based STE methods, which is attributed to the error accumulation in the denoising process [27] as well as the domain gap between training and inference [19]. To overcome this limitation, we introduce a glyph-adaptive mutual self-attention mechanism to improve the generator, which sets up a parallel reconstruction branch to introduce the source image style prior through cross-branch integration. The refined sampling process effectively eliminates visual inconsistency without requiring additional tuning.

Additionally, the deficiency of real-world evaluation benchmarks on STE has become a non-negligible problem as increasing methods are proposed. Early assessments [1], which rely on synthetic data, face significant limitations in practice due to the domain gap. Recent evaluations [14] emphasize text accuracy in edited real images but fail to benchmark visual quality adequately. Based on the observation that scene texts often occur in phrases with the same style and background in real-world scenery, we elaborately collect 1,280 text image pairs in terms of similar style and word length from scene text datasets to build the **ScenePair** dataset enabling comprehensive evaluation.

In summary, we improve STE with the full leverage of **Text** prior for comprehensive guidance **Control** throughout the model design, network training and inference control, termed as **TextCtrl**. Our main contributions are as follows:

- For the first time, we decompose the prerequisite of STE into fine-grained style disentanglement as well as glyph structure representation and incorporate the Style-Structure guidance with diffusion models to improve rendering accuracy and style fidelity.

- For further style coherency control during sampling, with the leverage of additional prior guidance through the reconstruction of the source image, we introduce a glyph-adaptive mutual self-attention mechanism that effectively eliminates visual inconsistency.

- We propose an evaluation benchmark **ScenePair** consisting of cropped text image pairs along with original full-size images. To the best of our knowledge, it is the first pairwise real-world dataset for STE which enables both visual quality assessment and rendering accuracy evaluation.

## 2  Related work

**GAN-based Scene Text Editing.**   SRNet [1] first introduces the word-level editing method built in a divide-and-conquer manner. SwapText [13] further enhances SRNet with Thin Plate Spline Interpolation Network for curved text modification while STRIVE [28] extends the framework into the video domain of scene text replacement. Besides, TextStyleBrush [4] adopts a self-supervised strategy building on StyleGAN2 [29] while MOSTEL [14] designs a semi-supervised training scheme.

**Diffusion-based Scene Text Editing.**   Numerous studies have focused on adapting the diffusion model for scene text manipulation. DiffSTE [20] improves pre-trained diffusion models with a dual encoder design, wherein a character encoder for render accuracy and an instruction encoder for style control is used. DiffUTE [23] further utilizes an OCR-based image encoder as an alternative to CLIP Text encoder. Moreover, TextDiffuser [21] and UDiffText [30] leverage character segmentation masks for condition input and supervised labels respectively. To leverage text style, LEG [18] concats the source image as input while DBEST [19] relies on a fine-tuning process during inference. Recently, AnyText [22] adopted a universal framework to resolve STE and STG in multiple languages based on the prevalent ControlNet [31].

**Image Editing with Diffusion Models.**   Image editing aims to manipulate a certain attribute (e.g. color, posture, position) of the target object while keeping the other context unchanged, which can be seen as the parent task of STE. Recent Diffusion-based methods have shown unprecedented potential with a wide variety of designs. Model-tuning methods [32, 33] fine-tune the entire model to enhance subject embedding in the output domain. Leveraging DDIM inversion [34], prompt-tuning methods [27] turn to improve identity preservation by optimizing null-text prompts through classifier-free guidance sampling [35]. Recently, [36, 37] explored the self-attention layers in LDMs and demonstrated the rich semantic information preserved in queries, keys and values. Through the cross-frame substitute of keys and values of self-attention, they perform non-rigid editing without additional tuning. [38] further extends the cross-frame interaction to video domain for motion editing.

## 3  Method

Based on a conditional synthesis manner, in this work, we define the scene text editing process as $I_{edit} = \mathcal{G}(C_{struct}, C_{style})$ as shown in Fig. 2 (c). The text glyph structure feature is acquired from a character-based structure encoder as $C_{struct} = \mathcal{T}(C_{text})$ in Fig. 2 (a) and the text style feature is derived from a style encoder $C_{style} = \mathcal{S}(I_{source})$ in Fig. 2 (b). The module design and pre-training strategy to enable precise extraction for glyph structure and fine-grained disentanglement of text style are introduced in section 3.1 and section 3.2 respectively, with the whole model training process illustrated in section 3.3. Furthermore, details of the improved tuning-free inference control and the proposed glyph-adaptive mutual self-attention mechanism in Fig. 2 (d) are presented in section 3.4.

### 3.1  Text Glyph Structure Representation Pre-training

Distinctive from natural objects, scene text possesses a complicated non-convex structure, wherein a minor stroke discrepancy can significantly alter visual perception and lead to misinterpretation [39], thus presenting unique challenges to editing accuracy. For scene text editing, an ideal text encoder is capable of encoding the target text concerning glyph structure rather than semantic information

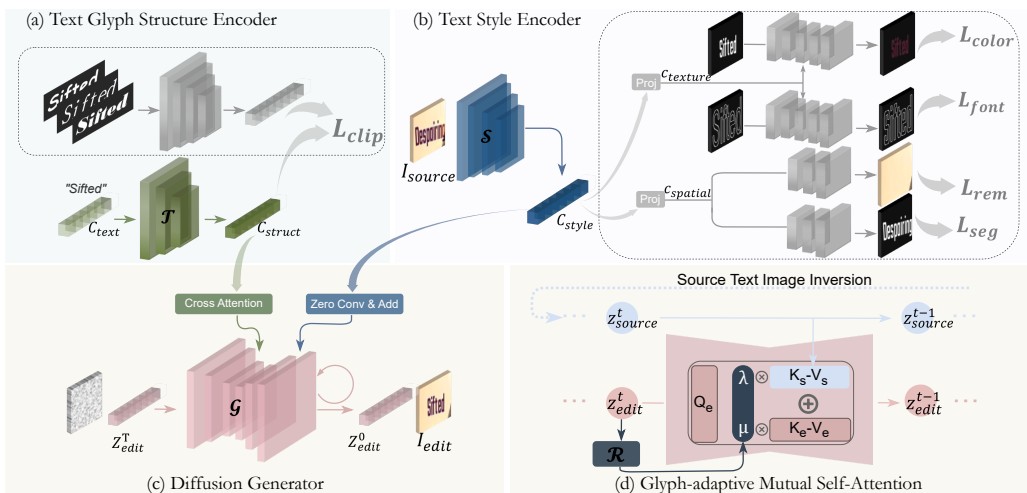

Figure 2: Decomposed framework of TextCtrl. (a) Text glyph structure encoder $\mathcal{T}$ with corresponding glyph structure representation pre-training. (b) Text style encoder $\mathcal{S}$ with corresponding style disentanglement pre-training. (c) Prior guided diffusion generator $\mathcal{G}$. (d) The improved inference control with the Glyph-adaptive Mutual Self-attention mechanism.

[20, 21] or certain image template [14, 23]. Specifically, for a certain text $C_{text} = "Sifted"$, encoder $\mathcal{T}$ is expected to be aware of the glyph structure of "$S$", "$i$", "$f$", "$t$", "$e$", "$d$" respectively.

To this end, we adopt a character-level text encoder to align the target text feature with its visual glyph structure. As depicted in Fig. 2 (a), the target text embedding in character level is processed with a transformer encoder $\mathcal{T}$ to generate glyph structure features $C_{struct} \in \mathbb{R}^{L \times d}$, which is further aligned to the visual feature of corresponding text image extracted by a frozen pre-trained scene text recognizer with CLIP loss [30, 40] $\mathcal{L}_{clip}$. Differ from [30], we collect vast quantities of text fonts constructing a cluster $\{font_1, font_2...font_n\}$ to render the corresponding text image with diverse fonts during training. The font-variance augmentation brings continuous glyph structure variation which implicitly enhances the projection from $C_{struct}$ to the cluster centroids of visual features for robust text glyph structure representation.

## 3.2 Text Style Disentanglement Pre-training

Text styles comprise a variety of aspects, including font, color, spatial transformation and stereoscopic effect, which visually mingle with each other and bring obstacles to disentangle the style features precisely in previous works. To realize the fine-grained disentanglement of the text style, we propose a multi-task pre-training paradigm as illustrated in Fig. 2 (b), involving text color transfer, text font transfer, text removal and text segmentation.

Concretely, given a text image $I_{source} \in \mathbb{R}^{3 \times H \times W}$, we first extract the style feature $C_{style} \in \mathbb{R}^{N \times d}$ with a ViT [41] backbone $\mathcal{S}$, which is projected to texture feature $c_{texture} \in \mathbb{R}^{N \times d}$ and spatial feature $c_{spatial} \in \mathbb{R}^{N \times d}$ respectively. Subsequently, $c_{texture}$ is employed in text color transfer and text font transfer while $c_{spatial}$ is utilized for text removal and text segmentation.

**Text Color Transfer.** Since both intrinsic style and lighting conditions determine the text color, it is challenging to label or classify the holistic color. Instead, we refer to image style transfer and implicitly extract color through colorization training. A light-weight encoder-decoder $\mathcal{F}^c$ is built to provide colorization on a black and white text image $i_{in}^c \in \mathbb{R}^{1 \times h \times w}$ with an Adaptive Instance Normalization [42] $\mathcal{A}$ for source text color image $i_{out}^c \in \mathbb{R}^{3 \times h \times w}$ written as:

$$i_{out}^c = \mathcal{F}_{dec}^c(\mathcal{A}(\mathcal{F}_{enc}^c(i_{in}^c), c_{texture})), \tag{1}$$

**Text Font Transfer.** With a common intention with color transfer to capture stylized information but focusing on glyph boundary, font transfer is realized through the boundary reshaping process. Another

light-weight encoder-decoder $\mathcal{F}^f$ is employed to transfer a template font text glyph $i_{in}^f \in \mathbb{R}^{3 \times h \times w}$ to the source font text glyph $i_{out}^f \in \mathbb{R}^{3 \times h \times w}$ through Pyramid Pooling Module [43] $\mathcal{P}$ in latent space as:

$$i_{out}^f = \mathcal{F}_{dec}^f(\mathcal{P}(\mathcal{F}_{enc}^f(i_{in}^f), c_{texture})), \tag{2}$$

**Text Removal and Text Segmentation.** Text removal aims at erasing the text pixels and reasoning the background pixels covered by text while text segmentation decouples the spatial relationships between background and text. A residual convolution block with spatial attention mechanism [44] is adopted to construct a removal head $\mathcal{F}^r$ and a segmentation head $\mathcal{F}^s$ respectively to generate predicted background $i_{out}^r \in \mathbb{R}^{3 \times h \times w}$ and predicted mask $i_{out}^s \in \mathbb{R}^{1 \times h \times w}$ as:

$$i_{out}^r = \mathcal{F}^r(c_{spatial}), \quad i_{out}^s = \mathcal{F}^s(c_{spatial}), \tag{3}$$

**Multi-task Loss.** With the multi-task pre-training for fostering the text style extraction and disentanglement ability of $SE$, the whole loss function for style pre-training can be expressed as:

$$\mathcal{L}_{disentangle} = \mathcal{L}_{color}(i_{out}^c, i_{gt}^c) + \mathcal{L}_{font}(i_{out}^f, i_{gt}^f) + \mathcal{L}_{rem}(i_{out}^r, i_{gt}^r) + \mathcal{L}_{seg}(i_{out}^s, i_{gt}^s), \tag{4}$$

wherein we leverage MSE loss for $\mathcal{L}_{color}$, MAE loss for $\mathcal{L}_{rem}$ and Dice loss [45] for $\mathcal{L}_{font}$ and $\mathcal{L}_{seg}$. Synthetic groundtruth is leveraged for fine-grained supervision and the task-oriented pre-training achieves fine-grained textural and spatial disentanglement of stylized text images which fertilizes the style representation for downstream generator.

### 3.3 Prior Guided Generation

With the robust glyph structure representation $C_{struct}$ and fine-grained style disentanglement $C_{style}$ mentioned above, a diffusion generator $\mathcal{G}$ is employed to integrate the prior guidance and generate the edited result as shown in Fig. 2 (c). For $C_{struct}$, since the U-Net in latent diffusion models contains both self-attention and cross-attention, wherein the cross-attention focuses on the relation between latent and external conditions [16, 17], we replace the key-value in cross-attention modules of $\mathcal{G}$ with the linear projection of $C_{struct}$ to provide glyph guidance for improving accurate text rendering. For $C_{style}$, promising results have been shown by additional control injection [31] through the decoder of U-Net, based on which we apply the multi-scale style feature $C_{style}$ to the skip-connections and middle block of the model $\mathcal{G}$ to provide a style reference for high-fidelity rendering.

With the leverage of pre-trained model [17], the training is performed under a combined supervision on the synthetic text image data. Please refer to Appendix A for preliminaries of the diffusion model and Appendix B.2 for implementation details of training.

### 3.4 Inference Control

During inference of the diffusion-based STE model, undesired color deviation and texture degradation occasionally occur. Such discrepancy can be partly attributed to the error accumulation during the iterative sampling process [27, 36]. Besides, the domain gap between training and inference impedes style consistency in complicated real-world scenery. To control the visual style consistency, we attempt to ameliorate the inference process by injecting style prior from the source image into editing. Specifically, we propose the Glyph-adaptive Mutual Self-Attention mechanism, which seamlessly incorporates the style of source images throughout the deconstruction process.

**Reconstruction Branch.** Rather than transforming random noise samples into an image, our objective is to execute an image-to-image translation, ensuring the preservation of style features. Initially, we perform DDIM inversion [27, 46] to generate an initial latent $z_{source}^T$ from the source image $I_{source}$. The deconstructed inversion process enables a reconstruction branch $(z_{source}^T, z_{source}^{T-1}...z_{source}^0)$ of the source image parallel to the editing branch $(z_{edit}^T, z_{edit}^{T-1}...z_{edit}^0)$, which benefits the proposed integration process, as shown by the arrow in Fig. 2 (d).

**Glyph-adaptive Mutual Self-Attention Mechanism (GaMuSa).** Diverge from the general image editing, the target text modification of STE can lead to significant changes in the condition representation, which impedes text style preservation through general prompt-tuning methods [27].

**Algorithm 1** Glyph-adaptive Mutual Self-attention
___

**Input**: Inversion latent $z_{source}^T$, reconstruction condition embedding $c_{source}$, editing condition embedding $c_{edit}$ and target text embedding $emb_y$.
**Parameters**: Time step $t$, interval $\tau$, intensity parameter $\lambda$ and $\mu$.
**Output**: Denoised latent $z_{source}^0$ and $z_{edit}^0$.
___

1:  $t = T, \lambda = 0, \mu = 1, \tau = 5$;
2:  **for** $t = T, T-1...1$ **do**
3:  ─────────────────── *Reconstruction Branch*───────────────────
4:      $z_{source}^{t-1}, \{K_s, V_s\} \leftarrow \hat{\mathcal{G}}(t, z_{source}^t, c_{source})$;         ▷ *Self-Attention.*
5:  ───────────────────── *Editing Branch*─────────────────────
6:      **if** at intervals of $\tau$ **then**
7:         $emb_{edit} = \mathcal{R}(\mathcal{E}_{dec}(z_{edit}^t))$;
8:         $\lambda = (emb_{edit} \cdot emb_y)/(\|emb_{edit}\| * \|emb_y\|)$;      ▷ *Cosine Similarity.*
9:      **end if**
10:     $\mu = 1 - \lambda$;
11:     $\{K_{es}, V_{es}\} = \lambda\{K_s, V_s\} + \mu\{K_e, V_e\}$;         ▷ *Integration.*
12:     $z_{edit}^{t-1} \leftarrow \hat{\mathcal{G}}(t, z_{edit}^t, c_{edit}; \{K_{es}, V_{es}\})$;    ▷ *Glyph-adaptive Mutual Self-attention.*
13: **end for**
**Return** $z_{source}^0, z_{edit}^0$
___

Recently, [36, 37] have demonstrated that self-attention layers in the diffusion model focus on latent internal relations. Based on the characteristic, we perform a mutual self-attention process between two branches as shown in Fig. 2(d), wherein at denoising step $t$, the Key-Value features $\{K_s, V_s\}$ from reconstruction branch are introduced to the self-attention operation in editing branch. Rather than a direct replacement in the previous method [36], however, we prefer an integration of $\{K_s, V_s\}$ and $\{K_e, V_e\}$ to mitigate the domain gap between $z_{source}^t$ and $z_{edit}^t$.

It is worth noting that the original mutual self-attention process is hyper-parameter-sensitive to the starting time step. Premature initialization may introduce ambiguity and lead to spelling errors, whereas late intervention might fail to provide sufficient guidance. Inspired by the gradual deformation of text glyph during the iteration, we design a glyph-adaptive strategy to control the intensity of integration. Specifically, we employed a vision encoder $\mathcal{R}$ of the pre-trained text recognizer [47] to construct a glyph-adaptive strategy for the harmonious integration of Key-Value between branches. Specifically, during the iterative process, the intermediate latent $z_{edit}^t$ is decoded and processed with $\mathcal{R}$ for the cosine similarity calculation with the target text embedding $emb_y$ at intervals of $\tau$ steps to denote the glyph similarity of the intermediate edited image with target text. The similarity will serve as the intensity parameter $\lambda$ and $\mu$ for controlling the integration between $\{K_s, V_s\}$ from reconstruction branch and $\{K_e, V_e\}$ from editing branch. The result of integration $\{K_{es}, V_{es}\}$ is subsequently leveraged in the self-attention modules of the editing branch for style coherency enhancement. The overall sampling pipeline is illustrated in Alg. 1.

$$\text{GaMuSa} = \text{Softmax}(\frac{Q_e \cdot (\lambda K_s + \mu K_e)^\mathsf{T}}{\sqrt{d}}) \cdot (\lambda V_s + \mu V_e), \quad \mu = 1 - \lambda. \tag{5}$$

## 4 Experiments

### 4.1 Dataset and Metrics

**Training Data.** Based on [1, 3], we synthesize 200k paired text images for style disentanglement pre-training and supervised training of TextCtrl, wherein each paired images are rendered with the same styles (i.e. font, size, colour, spatial transformation and background) and different texts, along with the corresponding segmentation mask and background image. Furthermore, a total of 730 fonts are employed to synthesize the visual text images in text glyph structure pre-training.

**ScenePair Benchmark.** To provide assessments on both visual quality and rendering accuracy, we propose the first real-world image-pair dataset in STE. Specifically, we collect 1,280 image pairs with

| Metrics / Methods | ScenePair (Cropped Text Image) | | | | ScenePair (Full-size Image) | |
|---|---|---|---|---|---|---|
| | SSIM $\uparrow$ ($\times 10^{-2}$) | PSNR $\uparrow$ | MSE $\downarrow$ ($\times 10^{-2}$) | FID $\downarrow$ | SSIM $\uparrow$ ($\times 10^{-2}$) | FID $\downarrow$ |
| SRNet [1] | 26.66 $\pm$ 0.00 | 14.08 $\pm$ 0.00 | 5.61 $\pm$ 0.00 | 49.22 $\pm$ 0.00 | 98.91 | 1.48 |
| MOSTEL [14] | 27.45 $\pm$ 0.00 | 14.46 $\pm$ 0.00 | 5.19 $\pm$ 0.00 | 49.19 $\pm$ 0.00 | 98.96 | 1.49 |
| DiffSTE [20] | 26.85 $\pm$ 0.08 | 13.44 $\pm$ 0.04 | 6.11 $\pm$ 0.04 | 120.34 $\pm$ 1.52 | 98.86 (76.91) | 2.37 (96.78) |
| TextDiffuser [21] | 27.02 $\pm$ 0.11 | 13.96 $\pm$ 0.03 | 5.75 $\pm$ 0.05 | 57.01 $\pm$ 0.44 | 98.97 (92.76) | 1.65 (12.23) |
| AnyText [22] | 30.73 $\pm$ 0.55 | 13.66 $\pm$ 0.07 | 6.19 $\pm$ 0.14 | 51.79 $\pm$ 0.35 | 98.99 (82.57) | 1.93 (16.92) |
| TextCtrl | **37.56** $\pm$ 0.32 | **14.99** $\pm$ 0.15 | **4.47** $\pm$ 0.15 | **43.78** $\pm$ 0.17 | **99.07** | **1.17** |

Table 1: Text style fidelity assessment within text image level and full-size image level, highlighted with **best** and second best results. For full-size image evaluation, we replace the unedited region with the origin image while values in "()" denote the direct output of inpainting-based methods.

| Metrics / Methods | ScenePair | | ScenePair (Random) | | TamperScene [14] | |
|---|---|---|---|---|---|---|
| | ACC(%) $\uparrow$ | NED $\uparrow$ | ACC(%) $\uparrow$ | NED $\uparrow$ | ACC(%) $\uparrow$ | NED $\uparrow$ |
| SRNet [1] | 17.84 $\pm$ 0.00 | 0.478 $\pm$ 0.000 | 9.61 $\pm$ 0.00 | 0.422 $\pm$ 0.000 | 39.96 $\pm$ 0.00 | 0.776 $\pm$ 0.000 |
| MOSTEL [14] | 37.69 $\pm$ 0.00 | 0.557 $\pm$ 0.000 | 22.50 $\pm$ 0.00 | 0.451 $\pm$ 0.000 | **76.79** $\pm$ 0.00 | 0.858 $\pm$ 0.000 |
| DiffSTE [20] | 31.35 $\pm$ 0.35 | 0.538 $\pm$ 0.002 | 21.56 $\pm$ 0.69 | 0.487 $\pm$ 0.002 | - | - |
| TextDiffuser [21] | 51.48 $\pm$ 0.19 | 0.719 $\pm$ 0.003 | 33.99 $\pm$ 0.34 | 0.635 $\pm$ 0.004 | - | - |
| AnyText [22] | 51.12 $\pm$ 0.21 | 0.734 $\pm$ 0.005 | 25.05 $\pm$ 0.05 | 0.593 $\pm$ 0.003 | - | - |
| TextCtrl | **84.67** $\pm$ 0.34 | **0.936** $\pm$ 0.003 | **66.95** $\pm$ 0.13 | **0.869** $\pm$ 0.007 | 74.17 $\pm$ 0.55 | **0.909** $\pm$ 0.011 |

Table 2: Text rendering accuracy evaluation with different methods, highlighted with **best** and second best results. "Random" denotes that we replace the paired target text in SCENEPAIR with randomly chosen text to verify the model robustness. Note that we are not able to evaluate inpainting-based STE methods [20, 21, 22] on TamperScene [14] since it does not contain full-size images.

text labels from ICDAR 2013 [48], HierText [49] and MLT 2017 [50], wherein each pair consists of two cropped text images with similar text length, style and background, along with the original full-size images. Collecting methods and dataset details are introduced in Appendix C.

**Evaluation Dataset.** For a fair comparison, we conduct all the evaluations on real-world datasets. ScenePair consists of 1,280 cropped text image pairs along with original full-size images enabling both style fidelity assessment and text rendering accuracy evaluation. TamperScene [14] combines a total of 7,725 cropped text images with predefined target text to provide rendering accuracy evaluation. Nevertheless, it does not involve paired images for style assessment nor full-size images for evaluation on inpainting-based methods, demonstrating the necessity of the proposed ScenePair.

**Evaluation Metrics.** For visual quality assessment, we adopt the commonly used metrics including (i) *SSIM*, mean structural similarity; (ii) *PSNR*, the ratio of peak signal to noise; (iii) *MSE*, the mean squared error on pixel-level; (iv) *FID* [51], the statistical difference between feature vectors. For text rendering accuracy comparison, we measure with (i) *ACC*, word accuracy and (ii) *NED*, normalized edit distance, using an official text recognition algorithm [52] and corresponding checkpoint.

### 4.2 Performance Comparison

**Implementation.** We conduct the comparison of the proposed TextCtrl with two GAN-based methods: SRNet [1] and MOSTEL [14] as well as three diffusion-based methods: DiffSTE [20], TextDiffuser [21] and AnyText [22] with their provided checkpoints. The quantitative results are illustrated in Tab. 1 and Tab. 2 while the qualitative results for comparison are shown in Fig. 3 and Fig. 4. Notably, DiffSTE [20], TextDiffuser [21] and AnyText [22] conduct STE with an inpainting manner on a full-size image, for which we employed the corresponding full-size image of each pair in ScenePair with the target text area masked as input and crop the generated target text area for style evaluation on text image level. SRNet [1], MOSTEL [14] and TextCtrl resolve STE in a synthesis manner on a text image, for which we perform a perspective process to paste the generated image back to the full-size image for style evaluation on full image level.

**Text Style Fidelity.** The text style fidelity assessment is performed on both the text image level and the full image level of ScenePair to enable a comprehensive comparison among methods. On

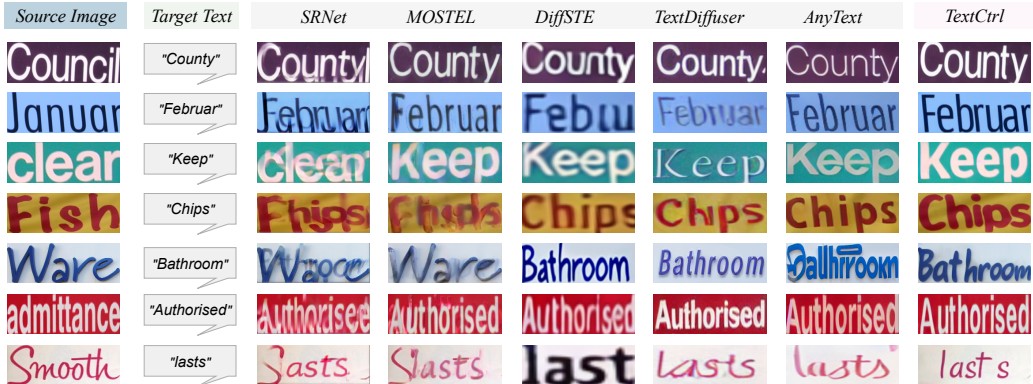

Figure 3: Qualitative comparison among different methods. Note that for the inpainting-based methods [20, 21, 22], we conduct the editing on the full-size images and perform the visualization of the edited text region.

text image level, TextCtrl outperforms other methods by at least 0.07, 0.53, 0.72 and 5.41 in SSIM, PSNR, MSE and FID respectively as represented in Tab. 1. GAN-based methods [1, 14] generally achieve a higher score on pixel-level assessment (i.e., PSNR, MSE). The reason lies in that they adopt a divide-and-conquer method, which contains a background restoration process to restrain the background region unaltered. Nevertheless, this may result in generating unsatisfied fuzzy images as shown in Fig. 3 column 3 and 4 due to the artifacts left by the unstable restoration process. Due to the loose style control result from the inpainting manner, undesired text style occurs occasionally in the outcome for diffusion-based methods [20, 21, 22]. On the contrary, TextCtrl benefits from the full leverage of disentangled style prior and the inference control for high-fidelity edited results. Further comparison on the full image level demonstrates the superiority of TextCtrl with precise manipulation and less style deviation against the inpainting-based methods with visualization in Fig. 4. Besides, it is not negligible that the inpainting strategy downgrades the image quality of unmasked regions.

**Text Rendering Accuracy.** Meanwhile, owing to the robust glyph structure representation, TextCtrl achieves superior spelling accuracy among all the methods, with more than 33% improvements in rendering accuracy of paired target text and randomly chosen text in ScenePair. TamperScene [14] contains a number of ambiguous low-resolution text images, which bring obstacles in style disentanglement and therefore impede the rendering accuracy of TextCtrl. Still, TextCtrl achieves a higher normalized edit distance that indicates the explicit mapping constructed between text and glyph. In comparison, GAN-based methods tend to yield ambiguous images, where source text left by imperfect removal blends with target text, resulting in fuzzy visual quality. Besides, due to the limited model capacity, GAN-based methods suffer from weak generalization and show incompetence with unseen style font as shown in Fig. 3 row 4 and 5. Diffusion-based methods achieve a disproportionate NED to their relatively low accuracy, which indicates their struggle with spelling mistakes. Notably, the inpainting manner serves as a primary factor that impedes the editing quality on small text for AnyText, whereas TextCtrl possesses the flexibility to perform editing on arbitrary scale text images.

### 4.3 Ablation Study

TextCtrl significantly improves STE through the substantial leverage of prior information in proposed (i) glyph structure representation pre-training, (ii) style disentanglement pre-training and (iii) glyph-adaptive mutual self-attention. We delve into the efficacy of each module in the following section.

**Text Glyph Structure Representation.** Conditional text prompt serves an important role in STE guiding the rendering of edited text glyphs. Consequently, we conduct the experiments by training TextCtrl with different text encoders $\mathcal{T}$ to evaluate the text rendering accuracy

| Text Encoder | ScenePair | | ScenePair (Random) | |
| --- | --- | --- | --- | --- |
| | ACC(%) ↑ | NED ↑ | ACC(%) ↑ | NED ↑ |
| CLIP [40] | 13.98 | 0.637 | 13.47 | 0.615 |
| $\mathcal{T}$ w/o font-variance | 76.08 | 0.875 | 60.84 | 0.827 |
| $\mathcal{T}$ w font-variance | **84.67** | **0.936** | **66.95** | **0.869** |

Table 3: Ablation experiment on glyph structure representation pre-training.

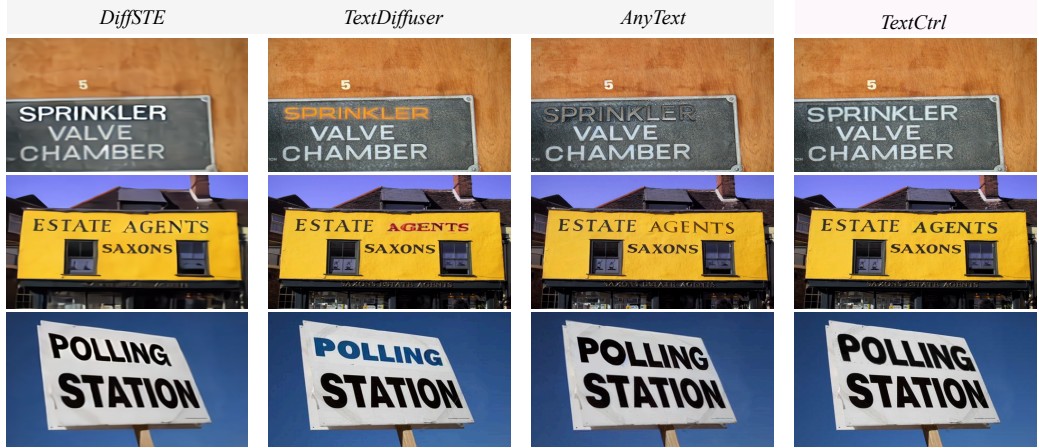

| DiffSTE | TextDiffuser | AnyText | TextCtrl |

Figure 4: Qualitative comparison with inpainting-based methods [20, 21, 22] on full-size images.

on ScenePair. Specifically, we employed a CLIP text encoder [40] for comparison which is generally adopted in generative diffusion models [16, 17]. The contrast between ACC and NED results in Tab. 3 confirms that the CLIP text encoder [40] struggles with spelling mistakes which are attributed to the sub-word embedding [24] and sub-optimal alignment between text prompt and text glyph. We further analyze the impact of the proposed font-variance alignment strategy in pre-training and the results indicate the robust representation brought by the augmentation.

**Text Style Disentanglement.** The explicit text style disentanglement pre-training distinguishes TextCtrl from previous STE methods [14, 21, 22] in fostering the fine-grained feature representation ability on scene text concerning font, color, glyph and background texture. To further verify the style disentangling ability of TextCtrl, as depicted in Fig. 5, we visualize the style feature embedding using t-SNE [53] with text images from ICDAR 2013 [48] encoded by the pre-trained style encoder $\mathcal{S}$. From a broad perspective, text images with a similar color cluster in different regions of feature space which indicate the style representation of the image entirety. From a micro perspective, as shown

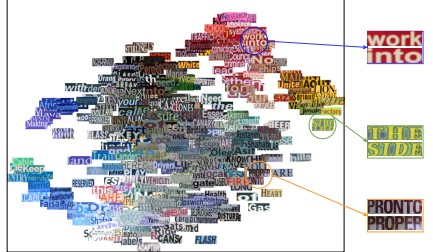

Figure 5: t-SNE [53] visualization of style features by pre-trained text style encoder.

by the sample pair pointed out with the magnifier, text images that share the same text style and background adjoin to each other regardless of different text content.

In Tab. 4, we replace the style encoder with a prevalent module ControlNet following the implementation settings and empirical suggestions in [31]. Concretely, the style encoder is replaced with a vanilla Stable Diffusion encoder, serving as the ControlNet module. The module is initialized with the pre-trained weight of SD encoder. As a powerful technique in enhancing generation controllability, ControlNet is prevalently leveraged in image editing for style and structural control. The

| Injection Module | SSIM ↑ | MSE ↓ | FID ↓ |
|---|---|---|---|
| ControlNet [31] | 0.3306 | 0.0464 | 58.30 |
| $\mathcal{S}$ w/o pre-training | 0.3130 | 0.0475 | 66.10 |
| $\mathcal{S}$ w pre-training | **0.3756** | **0.0447** | **43.78** |

Table 4: Ablation experiment on style disentanglement.

simple yet effective design enables a more meticulous reference from conditional input (e.g., Canny Edge, Depth map), which is also verified through the ablation study against our style encoder $\mathcal{S}$ (w/o pre-training) shown in Paper Tab. 4. After performing the style pre-training, however, the style encoder achieves a superiority performance against ControlNet module. Quantitative results demonstrate the fine-grained representation ability brought by the explicit style pre-training strategy, compared with implicit style learning by ControlNet. Notably, it also improves the parameter efficiency with 118M for style encoder $\mathcal{S}$ and 332M for ControlNet module.

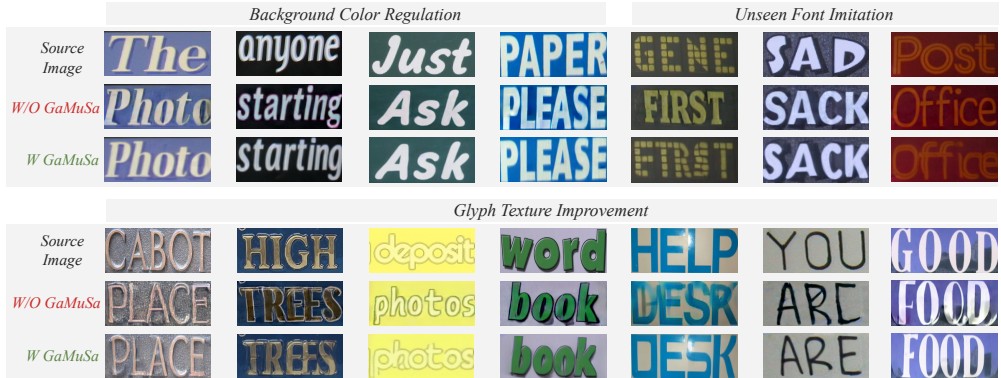

Figure 6: Visualization of text editing result with and without the proposed GaMuSa during inference, which verifies the improvements in background color, text font and glyph texture.

**Inference control with Glyph-adaptive Mutual Self-attention (GaMuSa).** In Tab. 5, we assess the style fidelity enhancement of GaMuSa in contrast with direct sampling and a prevalent enhancement method MasaCtrl [36] on ScenePair. Quantitative results verify the effectiveness of GaMuSa in enhancing style control during inference on variant real-world text images. Further visualization in Fig. 6 demonstrates the ability to persist style fidelity of GaMuSa when confronted with situations including background color deviation, unseen font and glyph texture degradation.

| Inference | SSIM ↑ | MSE ↓ | FID ↓ |
|---|---|---|---|
| w/o | 0.3126 | 0.0609 | 51.35 |
| w MasaCtrl [36] | 0.3571 | 0.0468 | 49.53 |
| w GaMuSa | **0.3756** | **0.0447** | **43.78** |

Table 5: Ablation experiment on inference enhancement.

## 5    Limitations and Conclusion

**Challenging arbitrary shape text editing.** Arbitrary shape text editing occurs occasionally when editing with text on a crescent signboard or a circular icon as shown in Appendix Fig. 11. These texts possess a complicated geometric attribution which is hard to disentangle through the style reference. Early works [13, 14] adopt the Thin-Plate-Spline (TPS) module to capture the accurate geometric distribution of text and perform a transformation on the template image as pre-processing. However, this strategy only takes effect in GAN-based methods which adopt an image-to-image paradigm. It remains a problem to effectively introduce accurate geometric prior guidance to diffusion models.

**Sub-optimal visual quality assessments metric.** Following previous STE methods, we adopt a variety of evaluation metrics for visual quality assessment. However, these metrics either focus on pixel-level discrepancy or concentrate on feature similarity in latent space, which is sub-optimal for assessing text style coherency. Besides, all these metrics rely on the paired data under which a ground-truth image is required. Though we collect a real-world image-pair dataset ScenePair in our work, a large amount of real-world text images remain unpaired and thus fail to provide visual quality assessment in editing. While human evaluation may be a possible solution, a more efficient and objective visual assessment metric is expected for scene text editing.

In this paper, we propose a diffusion-based STE method named TextCtrl with the leverage of disentangled text style features and robust glyph structure guidance for high-fidelity text editing. For further coherency control during inference, a glyph-adaptive mutual self-attention mechanism is introduced along with the parallel sampling process. Additionally, an image-pair dataset termed ScenePair is collected to enable the comprehensive assessment on real-world images. Extensive quantitative experiments and qualitative results validate the superiority of TextCtrl.

## Acknowledgments

This work is supported by the National Natural Science Foundation of China (Grant NO 62376266), and by the Key Research Program of Frontier Sciences, CAS (Grant NO ZDBS-LY-7024).

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

# Appendix

## A  Preliminaries

**Latent Diffusion Models (LDMs).**  Instead of operating diffusion process [34, 54] in image pixel space, LDMs utilize an autoencoder [55] $\varepsilon$ to translate the input image $x$ to latent space representation $z_0$. Then the denoising network $\epsilon_\theta$ built upon a time-conditional UNet [56] is trained to estimate added noise $\epsilon$ at a time step $t$. The condition embedding $c$, which is $c = \{C_{struct}, C_{style}\}$ in our work, is integrated through the cross-attention mechanism or the skip-connections and middle block, realizing a conditional generation that has the following training objective:

$$\mathcal{L}_{dn} = \mathbb{E}_{\varepsilon(x), c, \epsilon \sim \mathcal{N}(0,1), t}\big[\|\epsilon - \epsilon_\theta(t, z_t, c)\|_2^2\big], \tag{6}$$

**DDIM Sampling.**  At inference, random Gaussian noised $z_T$ can be gradually denoised to form a result $z_0$ through iterative sampling, wherein the deterministic DDIM sampling [34] is adopted in our method:

$$z_{t-1} = \sqrt{\frac{\alpha_{t-1}}{\alpha_t}} z_t + \sqrt{\alpha_{t-1}}\left(\sqrt{\frac{1}{\alpha_{t-1}} - 1} - \sqrt{\frac{1}{\alpha_t} - 1}\right) \cdot \epsilon_\theta(t, z_t, c), \tag{7}$$

where $\alpha_t = \prod_{i=1}^{t}(1 - \beta_i)$ and $\beta_0 = 0$ and tends to 1 as i increases.

**DDIM Inversion.**  In contrast to the stochastic sampling employed in DDPM [54], the sampling process in DDIM [34] is deterministic, allowing for the complete inversion [27, 46] from the original images latent $z_0$ back to initial noised latent $z_T$ to construct the reconstruction branch in our work.

$$z_{t+1} = \sqrt{\frac{\alpha_{t+1}}{\alpha_t}} z_t + \sqrt{\alpha_{t+1}}\left(\sqrt{\frac{1}{\alpha_{t+1}} - 1} - \sqrt{\frac{1}{\alpha_t} - 1}\right) \cdot \epsilon_\theta(t, z_t, c), \tag{8}$$

**Classifier-Free Guidance (CFG).**  To improve the visual quality and faithfulness of generated images, [35] introduces the classifier-free guidance technique, which jointly trains a conditional and an unconditional (denoted as $c_{null}$) diffusion model to provide a refined result:

$$\hat{\epsilon}_\theta(t, z_t, c) = \omega \cdot \epsilon_\theta(t, z_t, c) + (1 - \omega) \cdot \epsilon_\theta(t, z_t, c_{null}), \tag{9}$$

where $\omega$ is a hyperparameter that controls the strength of guidance.

## B  Implementation Setting

### B.1  Details of Model Architecture and Parameter

TextCtrl primarily comprises five components: an Encoder-Decoder VAE $\mathcal{E}$, a U-Net backbone $\mathcal{G}$, a text glyph structure encoder $\mathcal{T}$, a text style encoder $\mathcal{S}$ and a vision encoder $\mathcal{R}$. For the VAE and U-Net, we employ the pre-trained checkpoint of Stable Diffusion [17] V1-5[2]. For the text glyph structure encoder, we utilize a lightweight transformer encoder and perform pre-training on the proposed glyph structure representation aligning to the visual features captured by a frozen vision encoder[3] [57]. For the text style encoder, a ViT [58] backbone is employed to perform pre-training on multi-task style disentanglement. For the vision encoder, the vision backbone of ABINet[4] [47] is adopted with pre-trained checkpoint.

As the model input, the source image is resized to $256 \times 256$ while the max target prompt length is set to 24. The training process utilizes a batch size of 256 with a learning rate of $1 \times 10^{-5}$ and a total epoch of 100. TextCtrl is trained on 4 NVIDIA A6000 GPU and the parameter sizes of each module are provided in Tab. 6.

---

[2]https://huggingface.co/runwayml/stable-diffusion-v1-5
[3]https://github.com/roatienza/deep-text-recognition-benchmark
[4]https://github.com/FangShancheng/ABINet

| Modules | $\mathcal{G}$ | $\mathcal{E}$ | $\mathcal{T}$ | $\mathcal{S}$ | $\mathcal{R}$ | Total |
|---------|------|-----|-----|------|-----|-------|
| Params | 859M | 83M | 66M | 118M | 90M | 1216M |

Table 6: The parameter sizes of each module in TEXTCTRL

## B.2 Details of Training

During training of the generator, we follow the Diffusion Denoising Probabilistic Models (DDPM) [54] and perform the forward noising process on the image latent $z_0^e = \mathcal{E}_{enc}(I_e)$ with random $t \in \{1, ..., T\}$ to generate noised latent $z_t^e$:

$$z_t^e = \sqrt{\alpha_t} z_0^e + \sqrt{1 - \alpha_t} \epsilon_t, \epsilon_t \sim \mathcal{N}(0, I), \tag{10}$$

where $\alpha_t = \prod_{i=1}^t (1 - \beta_t)$ and $\beta_t \in (0, 1)$ is defined by a parameter schedule. The noise latent serves as the model input yielding the predicted noise $\tilde{\epsilon}_t$ along with condition embedding $c = \{C_{struct}, C_{style}\}$ as:

$$\tilde{\epsilon}_t = \mathcal{G}(t, z_t^e, c), \tag{11}$$

during which $c$ is randomly replaced by the unconditional embedding $c_{null} = \{C_{null}, C_{style}\}$ with the probability $p_{uc} = 0.1$ to jointly train an unconditional model, enabling the leverage of classifier-free guidance [35].

Since the diffusion process is performed in the latent space, to enable further vision and linguistics supervision, we construct the image-level result $\tilde{I}_e = \mathcal{E}_{dec}(\tilde{z}_e^0)$ where $\tilde{z}_e^0$ is gained through:

$$\tilde{z}_0^e = \frac{1}{\sqrt{\alpha_t}}(z_t^e - \sqrt{1 - \alpha_t} \tilde{\epsilon}_t), \tag{12}$$

We provide a triple-guidance loss for supervision of the diffusion generator, including the denoising loss $\mathcal{L}_{dn}$, the construction loss $\mathcal{L}_{cons}$ and the linguistic loss $\mathcal{L}_{ocr}$.

For construction loss, we adopt the $\mathcal{L}_{per}$ and $\mathcal{L}_{style}$ as part of construction loss following [14], along with MSE loss $\mathcal{L}_{reg}$. The functions are written as:

$$\mathcal{L}_{cons} = \lambda_1 \mathcal{L}_{per} + \lambda_2 \mathcal{L}_{style} + \mathcal{L}_{reg}, \tag{13}$$

$$\mathcal{L}_{per} = \mathbb{E}[\|\phi_i(I_e) - \phi_i(\tilde{I}_e)\|_1], \tag{14}$$

$$\mathcal{L}_{style} = \mathbb{E}_j[\|G_j^\phi(I_e) - G_j^\phi(\tilde{I}_e)\|_1], \tag{15}$$

where balance factors $\lambda_1$ and $\lambda_2$ are set to 0.01 and 100 respectively. $\phi_i$ is the activation map from *relu1_1* and *relu5_1* layer of VGG-19 model [59] and G is the Gram matrix.

Words are composed of character sequences that inherently contain linguistic information. pre-trained text recognition models, rich in sequence features prior, can be harnessed as global guidance to enhance text rendering ability. For linguistic loss, a recognition process will be applied to the decoded image $\tilde{I}_e \in \mathbb{R}^{3 \times H \times W}$ generating recognition result $\tilde{y}$, which is utilized to calculate the cross-entropy loss $CE$ with the text label $y$:

$$\mathcal{L}_{ocr} = \lambda_3 CE(y, \tilde{y}), \tag{16}$$

where $\lambda_3$ is set to 0.01. The overall objective function in training can be presented as a combination of the denoising loss Eq. 6, the construction loss Eq. 13 and the text recognition loss Eq. 16:

$$\mathcal{L} = \mathcal{L}_{dn} + \mathcal{L}_{cons} + \mathcal{L}_{ocr}. \tag{17}$$

## B.3 Details of Inference

During inference, for a full-size image, the quadrangle location of the source text region is indicated through user interface or detection with ocr model, which is the same as all the other STE methods to provide clear instruction of the specific text to be edited. Subsequently, we perform a cropping and perspective process on the text region to acquire the input source image. The source image is further employed in an inversion process based on Eq. 8 to construct a reconstruction branch and disentangled in style encoder $\mathcal{S}$ to serve as style guidance $C_{style}$, while the arbitrary target text provided by user is encoder by glyph structure encoder $\mathcal{T}$ as $C_{struct}$. The condition embedding

$c = \{C_{style}, C_{strcut}\}$ is later passed to the generator $\mathcal{G}$ guiding the generation process. The sampling step is set to $T = 50$ and the classifier-free guidance scale is set to $\omega = 2$ with 7 seconds to generate an edited image on a single NVIDIA A6000 GPU. Note that for the reconstruction branch, either a default null text or a source text is suitable for the input of glyph structure encoder $\mathcal{T}$ due to the symmetry of the inversion process Eq. 8 and the reconstruction process Eq. 7, which is flexible and will not interface the editing quality. The edited result will be perspective and stitched back to the original region for the full image result.

## C Sic ScenePair Dataset

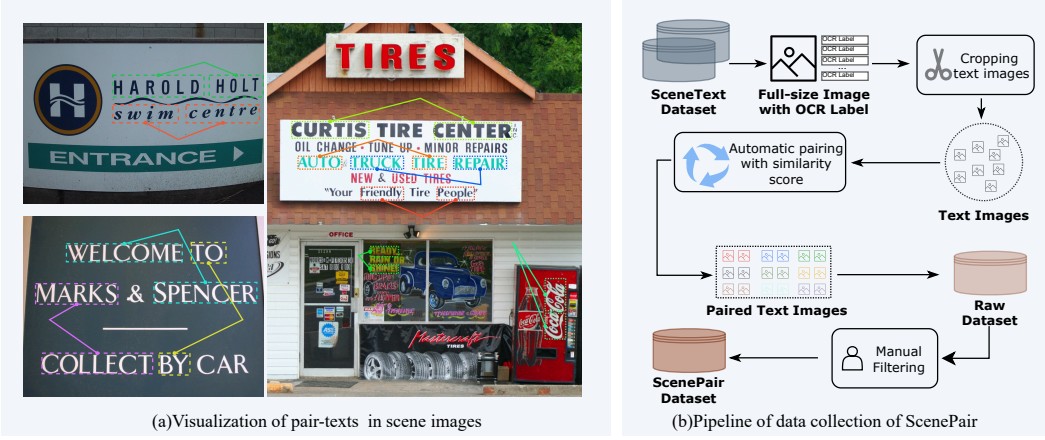

(a)Visualization of pair-texts in scene images    (b)Pipeline of data collection of ScenePair

Figure 7: Data collection strategy of ScenePair. (a) Texts with the same style and background often occur in real-world scenery. (b) The pipeline for our data collection.

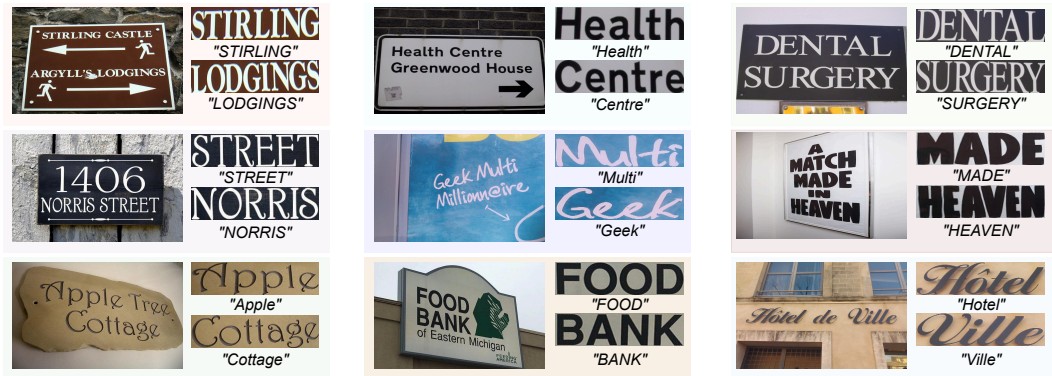

Figure 8: Visualization of paired data along with full-size images in SCENEPAIR.

**Dataset Introduction.** To provide practical assessments on both style consistency and rendering accuracy, we propose the first real-world image-pair dataset in STE termed ScenePair. Specifically, we collect 1,280 image pairs with the text label from ICDAR 2013 [48], HierText [49] and MLT 2017 [50]. For each pair, we collect the source text image, the target text image, the respective text labels, the respective quadrangle locations in full-size image and the original full-size image.

The inspiration for constructing ScenePair dataset comes from the observation that scene texts often occur in phrases with the same style and background in real-world scenery, as depicted in Fig. 7 (a), which serves as a perfect pair sample for evaluation of editing quality. Notably, there isn't a "correct" result for image editing whereas our paired data serves as a reference benchmark for high-fidelity.

**Collecting Strategy.** We design a semi-automatic collecting strategy for ScenePair from several scene text datasets as illustrated in Fig. 7 (b). Initially, we collect the full-size images from the

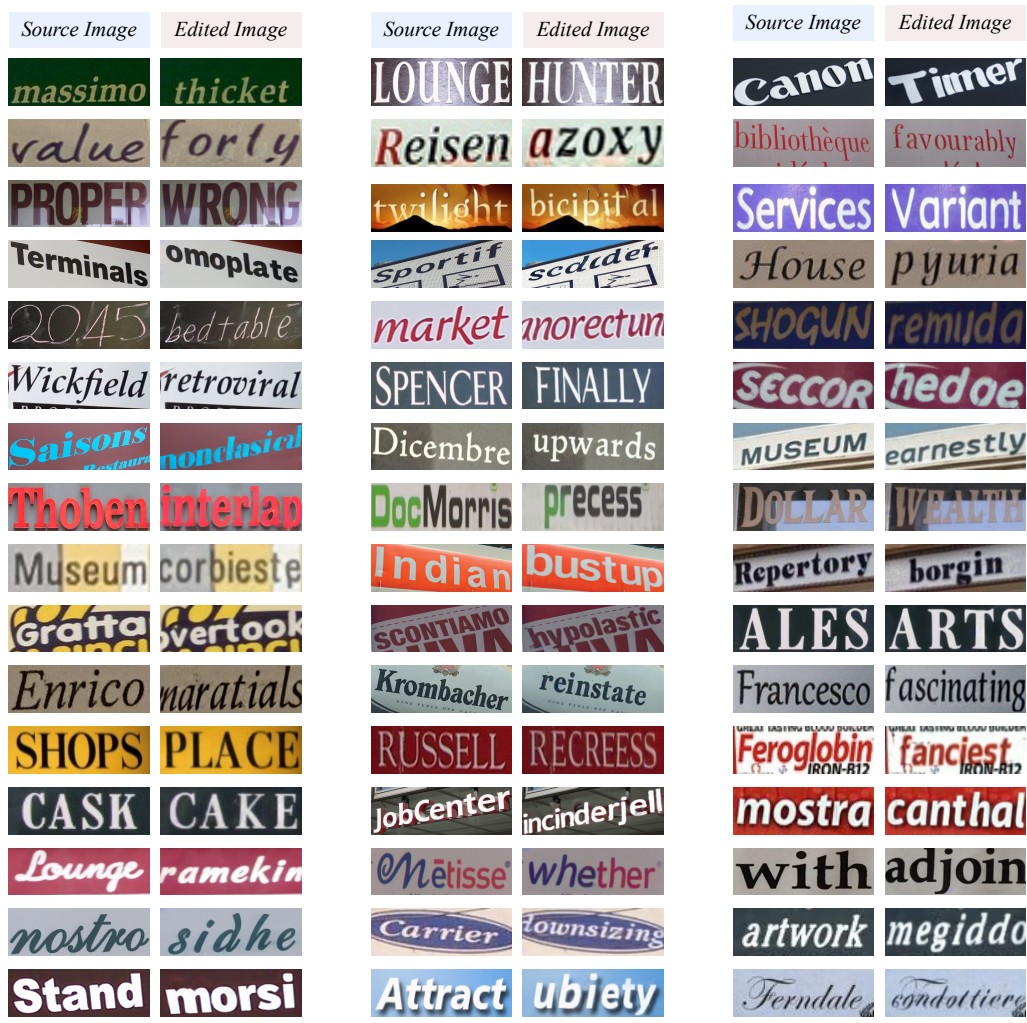

Figure 9: Visualization of edited result on text image by TEXTCTRL on dataset TAMPERSCENE.

datasets along with the detection and recognition labels. For each full-size image, we perform cropping and perspective with the quadrangle location to acquire all the text images, for which an automatic pairing algorithm is employed to construct paired text images. Specifically, the algorithm calculates the weighted similarity score involving text length, aspect ratio, centre distance in full-size image and SSIM (Structure Similarity Index Measure) between the cropped text images, wherein the paired images with similarity score that surpass a pre-defined threshold would be chosen to form a raw dataset. Finally, we manually filter out the unsatisfied pairs to construct the ScenePair dataset, as shown in Fig. 8.

# D   Visualization

To further demonstrate the editing performance of our TextCtrl, we provide various visualizations of edited outcomes. In Fig. 9, a variety of images in TamperScene with abundant text style are provided to verify the text style generalization of TextCtrl. In Fig. 10, we provide the edited result on the scene image of ICDAR 2013. Specifically, given a scene photo with an automatically detected text box, we perform scene text editing on box images and stitch back to the original photo. Results demonstrate the ability of TextCtrl to preserve fine-grained detail for high-fidelity editing.

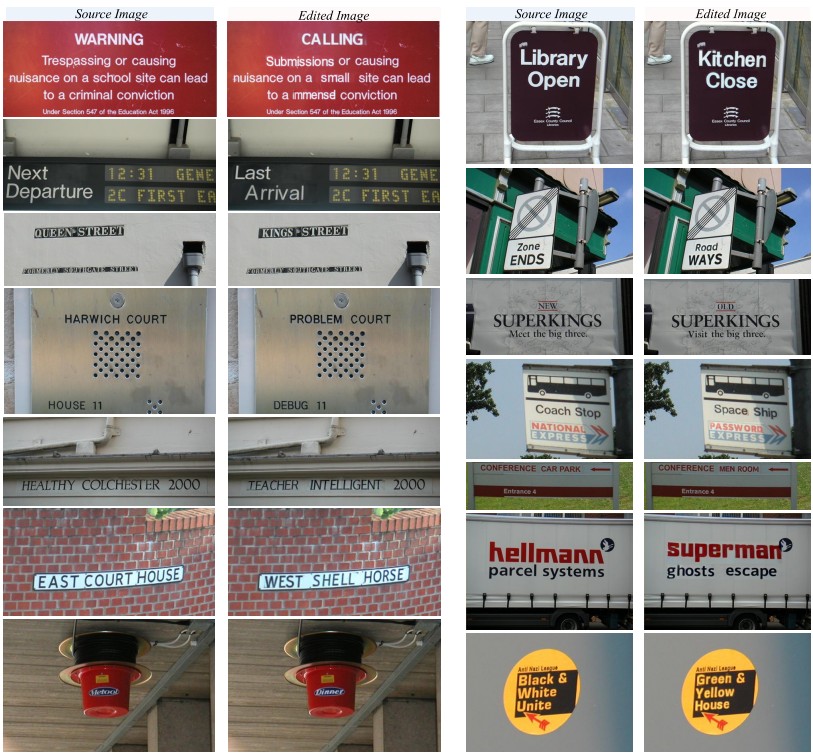

Figure 10: Visualization of edited result on scene image by TEXTCTRL on dataset ICDAR 2013.

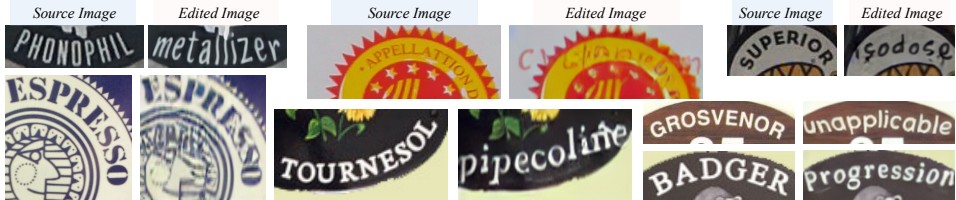

Figure 11: Visualization of failure cases on text image by TEXTCTRL. The problem mainly results from the insufficient geometric prior guidance control.

# E  Additional Considerations

This paper introduces a novel method for scene text editing which leverages the fine-grained style disentanglement and robust glyph structure representation to achieve high-fidelity editing results. Though we acknowledge that the proposed method has the potential to be misused for image forgery, significant advancement in visual quality and text rendering accuracy would also contribute to text-related visual art creation. To prevent the high risk of misuse of the proposed method, an additional user commitment will be required for accessing the checkpoint in our forthcoming open release, through which we hope to alleviate the potential misuse while benefiting further research.

# F  Licenses

Here we provide license details of the code and data used in our proposed network and comparison experiments. SRNet [1] is available for use under GNU General Public License v3.0. AnyText [22], deep-text-recognition-benchmark [52], ViTSTR [57] are available for use under Apache License 2.0. DiffSTE [20], TextDiffuser [21], CLIP [40] are available for use under MIT License. Stable Diffusion [16] is available for research purposes under CreativeML Open RAIL M License. ABINet

is available for non-commercial purposes with license [5]. MLT 2017 [50] and HierText [49] are released under CC BY-SA 4.0 license. There is no known license for ICDAR 2013[6] [48], MOSTEL[7] [14] and TamperScene[8] [14], but the data and code are commonly referred to as "public", and so we interpret this to mean they are available for use under research purpose.

---

[5]https://github.com/FangShancheng/ABINet/blob/main/LICENSE

[6]https://rrc.cvc.uab.es/?ch=2

[7]https://github.com/qqqyd/MOSTEL

[8]https://github.com/qqqyd/MOSTEL

