# OpenReview forum: "TextCtrl: Diffusion-based Scene Text Editing with Prior Guidance Control"
_NeurIPS.cc/2024/Conference — NeurIPS 2024 spotlight_

### Official Review · Reviewer_25Re · 2024-07-09

**Soundness:** 4
**Presentation:** 3
**Contribution:** 3
**Rating:** 7
**Confidence:** 5

**Summary:**

Considering the limitations of current GAN-based and Diffusion-based Scene Text Editing (STE) methods, this paper introduces TextCtrl, a diffusion-based method that edits text with prior guidance control. Specifically, TextCtrl incorporates text style disentanglement and text glyph structure representation to control the style and structure of generated scene text images, respectively. During inference, a Glyph-adaptive Mutual Self-attention mechanism is designed to further enhance style consistency and visual quality. Additionally, a new real-world image-pair dataset called ScenePair is crated for fair evaluation. Experiments demonstrate that TextCtrl outperforms previous methods in terms of style fidelity and text accuracy.

**Strengths:**

1.	The paper is well written, and the presentation is clear
2.	In the field of STE, the proposed method is reasonable in targeting structural integrity and stylistic consistency. By 3 prior guidance control of text style disentanglement, text glyph structure representation and source image reconstruction, TextCtrl can generate edited image with both high style fidelity and high recognition accuracy, which is novel.
3.	The paper constructs the first real-world image-pair dataset for STE, which is contributed to the community.
4.	The experimental results show that the proposed method surpasses previous methods with large margins in most cases.

**Weaknesses:**

1.	Several hyperparameters are included in the TextCtrl. I am not sure is there any experience for users to set up these hyperparameters for different datasets.

2.	Some figures could be improved. For example, Figure 3 is somewhat not easy to follow; highlighting the pretrained parts and the trainable parts would make it clearer. Figure 6 appears chaotic, it may be better to choose fewer texts.

**Questions:**

1.	Can this method be used on Chinese scenes?

2.	The authors reference the way of structure condition insertion in ControlNet to feed style condition into TextCtrl, i.e., zero convolution and add, rather than structure condition. Could the authors explain this modification? BTW, could the authors introduce more experimental settings for ControlNet in Table 3, including inputs?

3.	In Fig.3, why using "sifted" for c_texture but "Despairing" for c_spatial?

**Limitations:**

Yes, the limitations are detailed discussed.

---

> ### Author Rebuttal · Authors · 2024-08-06
>
> Thank you for the detailed review. We are encouraged by the positive comments on novelty of the method and the contribution of the proposed dataset. The concerns are taken care to address point by point in the following.
>
>
> > **[W1]: Discussion about the hyperparameters in TextCtrl.**
>
> [A1]: Thanks for the question. Briefly, we have made an effort to reduce the demand for manually setting hyperparameters and aligned the rest to regular settings. Specifically,
>
> -  In pretraining, we revise the max-length for text structure encoder to 24 since it focuses on character-level representation of words rather than long context.
>
> -  For training and sampling, we follow the regular settings according to [*1*].
>
> -  For integration in inference, as shown in ***Paper Fig.3(d)*** and ***Paper Appendix Alg.1***, we set the initial intensity parameter $\lambda=0$ and $\mu=1$ to avoid the distraction of self-attention in early sampling stages, facilitating text structure initialization. Subsequently, $\lambda$ is automatically calculated by the structural similarity of intermediate latent and target text with $\mu=1-\lambda$, which allows an adaptive integration and avoids manual settings.
>
> Through aforementioned settings, the hyperparameters are determined and they are set consistently to perform stable and robust editing, irrespective of different datasets.
>
>
>
> > **[W2]: Figure 3 and figure 6 could be improved.**
>
> [A2]: Thanks for the suggestion on improving ***Paper Fig.3/6*** for a more intuitive presentation. Briefly, ***Paper Fig.3(a)(b)*** depicts the pretraining of text encoder *TE* and the style encoder *SE* respectively, while ***Paper Fig.3(c)(d)*** presents the training (with *TE* frozen) and inference of the whole framework. We will improve the figures in the revised paper.
>
>
>
> > **[Q1]: Can this method be used on Chinese scenes?**
>
> [A3]: Although the discussion in the paper mainly focuses on English scenes, our method is promising to realize editing on Chinese scenes by simply replacing the dataset. We highlight the effectiveness of the proposed text structure pretraining on aligning the character-level representation and the visual glyph structure, as well as the text style pretraining on disentangling text style. Since both modules are trained solely on synthetic data, they can be easily modified to fit in another language scenes. In addition, the proposed inference control effectively leverages the style information from real text image, which further enhances the model's adaptability to different scenes.
>
>
>
> > **[Q2]:The authors reference the way of structure condition insertion in ControlNet to feed style condition into TextCtrl, i.e., zero convolution and add, rather than structure condition. Could the authors explain this modification? BTW, could the authors introduce more experimental settings for ControlNet in Table 3, including inputs?**
>
> \[A4\]: Sure. We will first explain the reason for the modification and later detail the ControlNet in ***Paper Tab.3***.
>
> As stated in ***Paper Sec.3***, TextCtrl is built based on a synthesis manner, wherein text is the essential product of synthesis/generation while style (e.g., font, background) serves as the additional reference. As a result, text structure is introduced through cross-attention to enable the basic generation of text image while style is inserted through ''zero convolution and add'' in decoder for reference. In fact, rather than modified, the design of TextCtrl coincides with ControlNet [2] which introduces additional reference to a foundational generative model.
>
> In the ablation study of ControlNet, we follow the implementation settings and empirical suggestions in [*2*]. Concretely, the style encoder is replaced with a vanilla Stable Diffusion encoder, serving as the ControlNet module. The module is initialized with the pretrained weight of SD encoder and the input includes the style text image $I_{source}$, the noised latent $z^{t}$, timestep $t$ and text structure embedding $C_{struct}$. During training and sampling, $I_{source}$ is first encoded and integrated with $z^{t}$, which subsequently interacts with $t$ and $C_{struct}$ in the ControlNet module and finally inserts into diffusion generator through the ''zero convolution & add'' process.
>
>
>
> > **[Q3]: In Fig.3, why use "sifted" for c_texture but "Despairing" for c_spatial?**
>
> [A5]: Thanks for your detailed review of ***Paper Fig.3(b)***.
>
> Briefly, both $c_{texture}$ and $c_{spatial}$ are encoded from $I_{source}$ *("Despairing")*. The image of *"Sifted"* is rather integrated with $c_{texture}$ than used for $c_{texture}$ in respective task.
>
> Specifically, for an input $I_{source}$ *("Despairing")*,
>
> - $c_{spatial}$ is leveraged in *Text Removal* and *Text Segmentation* for capturing the spatial style. As a result, a pure background and a segmentation mask of *"Despairing"* are expected as the direct output.
>
> - $c_{texture}$ is leveraged in *Color Transfer* and *Font Transfer* for capturing the texture style. With $c_{texture}$ encoded from the stylized image of "Despairing", to avoid degeneration of the transfer model into an identity mapping network, we turn to a different text image *"Sifted"* (synthesized in the same color/font) as the sub-models input/output with $c_{texture}$ as conditional style guidance. The substitution enforces the disentanglement of color/font from the source image context and the implementation details are given in ***Paper Sec.3.2***.
>
> The aforementioned processes ensure the encoded features of style encoder and the objectives of each task are closely associated with the text style disentanglement ability.
>
> > **References**
>
> [1] Rombach et al. High-Resolution Image Synthesis with Latent Diffusion Models. CVPR, 2022.
>
> [2] Zhang et al. Adding Conditional Control to Text-to-Image Diffusion Models. ICCV, 2023.

---

> > ### Comment · Reviewer_25Re · 2024-08-14
> >
> > Thanks for the responses, which addressed my concerns. Hence, I increase my rating to "Accept". However, the authors should make hyperparameters setting clear, make Fig.3&6 easy to understand, and explain modification for ControlNet in the final version if it is accepted.

---

> > > ### Author Response · Authors · 2024-08-14
> > >
> > > Thank you for your feedback. We will work on refining the mentioned issues and enhancing clarity in the revised paper. Thanks again for the thorough review and valuable suggestions!

---

### Official Review · Reviewer_mqKG · 2024-07-11

**Soundness:** 4
**Presentation:** 3
**Contribution:** 4
**Rating:** 7
**Confidence:** 4

**Summary:**

This paper aims to enhance scene text editing performance using a conditional prior-guidance-control diffusion model. It decomposes text style into background, foreground, font glyph, and color features. A text glyph structure representation improves the correlation between the text prompt and glyph structure. For inference, a glyph-adaptive mutual self-attention mechanism with a parallel sampling process is proposed. To evaluate effectiveness, the ScenePair image-pair dataset is created. Experiments on ScenePair and TamperScene datasets, along with ablation studies, are conducted.

**Strengths:**

1) Text glyph structure representation addresses the weak correlation between text prompts and glyph structures in text-to-image diffusion models.

2) The proposed glyph-adaptive mutual self-attention ensures coherent control during inference, guided by source image reconstruction, which is novel.

3) The ScenePair dataset is valuable for research on STE tasks.

4) Both quantitative and qualitative experimental results demonstrate notable performance improvement across multiple datasets.

**Weaknesses:**

1) For the ablation study of text style disentanglement, why replace the style encoder with ControlNet?

2) Some sentences are too long which impacts the clarity. For example, line 235 to line 239, the sentence “GAN-based methods generally … due to the unstable restoration quality” expands 5 lines, and it will be more clear if it is split into two or more sentences.

3) Some figures and tables are not convenient for reading, such as Fig. 4 & 5, Tab. 4 & 5. They should be swapped so that the main parts are near the corresponding figure or table.

**Questions:**

See the weaknesses

**Limitations:**

The authors have pointed out the limitations in the Appendix.

---

> ### Author Rebuttal · Authors · 2024-08-06
>
> Thank you for the valuable comments. Your detailed review will certainly help improve the revised paper. The remaining concerns are taken care to address point by point in the following.
>
>
> > **[W1]: For the ablation study of text style disentanglement, why replace the style encoder with ControlNet?**
>
> [A1]: Thanks for bringing the question. We will first detail the experiment and later discuss the purpose.
>
> In ***Paper Tab.3***, the ablation study of ControlNet [*1*] is conducted by replacing our style encoder with a vanilla Stable Diffusion encoder, serving as the ControlNet module. Following the implementation settings and empirical suggestion in [*1*], we initialize the parameters of ControlNet module with the copy of pretrained SD encoder weight and enable training on both ControlNet module and SD generator to fit in the specialized text data.
>
> As a powerful technique in enhancing generation controllability, ControlNet is prevalently leveraged in image editing for style and structural control. The simple yet effective design enables a more meticulous reference from conditional input (e.g., Canny Edge, Depth map), which is also verified through the ablation study against our style encoder (w/o style pretraining) shown in ***Paper Tab.3***. After performing the style pretraining, however, the style encoder achieves a superiority performance against ControlNet module.
>
> The ablation study is leveraged to demonstrate the fine-grained representation ability brought by the explicit style pretraining strategy, compared with implicit style learning by ControlNet. Notably, it also improves the parameter efficiency with 118M for style encoder and 332M for ControlNet module.
>
>
>
> > **[W2]: Some sentences are too long which impacts the clarity. For example, line 235 to line 239, the sentence “GAN-based methods generally … due to the unstable restoration quality” expands 5 lines, and it will be more clear if it is split into two or more sentences.**
>
> [A2]: We appreciate your detailed review. In our revised paper, we will rewrite this kind of sentences with a clearer presentation.
>
> *"GAN-based methods generally achieve a higher score on pixel-level assessment (i.e., PSNR,MSE). The reason lies in that they adopt a divide-and-conquer method, which contains a background restoration process to restrain the background region unaltered. Nevertheless, this may result in generating unsatisfied fuzzy images as shown in Fig.5 column 3 and 4 due to the artifacts left by the unstable restoration process."*
>
>
>
> > **[W3]: Some figures and tables are not convenient for reading, such as Fig. 4 & 5, Tab. 4 & 5. They should be swapped so that the main parts are near the corresponding figure or table.**
>
> [A3]: Thanks for the suggestion. We will rearrange the order of the figures and tables in our revised paper for better readability. Please don't hesitate to let us know if you have further questions or suggestions.
>
> > **References**
>
> [1] Zhang et al. Adding Conditional Control to Text-to-Image Diffusion Models. ICCV, 2023.

---

> > ### Comment · Reviewer_mqKG · 2024-08-13
> > **post-rebuttal**
> >
> > After reviewing the feedback and rebuttal, I find that the concerns have been addressed. I will maintain my rating.

---

> > > ### Author Response · Authors · 2024-08-13
> > >
> > > Thank you for your feedback! We truly appreciate your time and effort in reviewing our paper. We are committed to continuous improvement and value your insights in the revised paper.

---

### Official Review · Reviewer_rTPt · 2024-07-12

**Soundness:** 3
**Presentation:** 3
**Contribution:** 2
**Rating:** 5
**Confidence:** 4

**Summary:**

This paper proposes TextCtrl, which is a new method for high-fidelity scene text editing. The authors identify the primary factor hindering previous methods from achieving accurate and faithful scene text editing to be the absence of prior guidance on stylistic elements and textual organization. By leveraging disentangled text style features and robust glyph structure guidance, TextCtrl achieves superior editing performance. A glyph-adaptive mechanism and a new ScenePair dataset for real-world evaluation further solidify TextCtrl's effectiveness.

**Strengths:**

- The paper is well-written.
- The experiments are extensive and well-discussed.
- The dataset is valuable for the research community.

**Weaknesses:**

- The paper misses details and discussions on key contributions (please see the Questions section below), making it hard to evaluate the significance of the contributions of the work.

**Questions:**

- Sec. 3.2 on text style disentanglement only provides implementation details but doesn’t give the big picture of why all these components working together would successfully learn to disentangle the text style. Since this seems to be one of the main contributions of the paper, I’d recommend to include more discussions on what each component achieves.
- Line 189 mentions integration of the K, V of the reconstruction branch and K, V of the main branch is preferred over the replacement that is often done in the literature. Could you provide some insights on why this might be more helpful, and in which scenarios?

**Limitations:**

The limitations have been thoroughly discussed in the appendix.

---

> ### Author Rebuttal · Authors · 2024-08-06
>
> Thank you for the constructive comments. We are encouraged by the positive response to the paper presentation and ScenePair dataset. The remaining concerns are taken care to address point by point in the following.
>
> > **[Q1]: A big picture of why all the components working together in style pretraining would successfully learn to disentangle the text style.**
>
> [A1]:
> Text style comprises a variety of aspects (e.g., font, color, perspective), which visually mingle with each other. This characteristic poses great challenge to the direct apply of general representation learning methods (e.g., vqvae [*1*]) on stylized text images. While it is intuitive to decompose text style into isolated components for object-oriented learning, some components (e.g., size, spatial relation) are comparatively difficult to depict or synthesize.
>
> Eventually, we turn to resolve the problem with task-oriented pretraining on the stylized text image, which effectively enables the leverage and representation of implicit text style during learning. Notably, we unify several tasks on the same model to promote mutual collaboration from joint training and enhance the style feature extraction, as suggested in [*2*].
>
> Four tasks are included in style pretraining as shown in ***Paper Fig.3(b)***, wherein
>
> - *(i) Text Color Transfer* extracts text color from the stylized text image. Since the text color is determined by both intrinsic style and lighting conditions, it is challenging to label or classify the holistic color (e.g., RGB, Captions). Instead, we refer to image style transfer and implicitly extract color through colorization training.
>
> - *(ii) Text Font Transfer* shares a common intention with *Text Color Transfer* to capture stylized information but focuses on glyph boundary that shapes the font style, for which we construct a similar architecture and learn from the boundary reshaping.
>
> - *(iii) Text Removal* has already been explored in considerable works [*3,4*], which aims at erasing the text pixels and reasoning the background pixels covered by text. We package it as one of the pretraining tasks to benefit the background preservation in editing.
>
> - *(iv)* *Text Segmentation* not only facilitates the precise removal but also decouples the spatial relationships between background and text, as well as extracting an internal description of text (e.g., size, interval of characters).
>
> The implementation details are provided in ***Paper Sec.3.2***. The task-oriented pretraining achieves fine-grained textural and spatial disentanglement of stylized text image and fertilizes the style representation for downstream generator.
>
>
>
>
> > **[Q2]: Why prefer the integration of K-V of reconstruction branch and main branch.**
>
> [A2]: This is indeed an interesting question concerning the semantic correspondence of diffusion latents.
>
> To start off, leveraging the internal representations, numerous works [*5,6*] have revealed the generative diffusion model's ability on establishing reasonable semantic correspondence across different images, even when exhibiting significant differences in categories, shapes, and poses.
>
> A step forward, [*7*] utilizes the aforementioned correspondence in self-attention module through a mask-guided replacement of K-V to perform non-rigid editing of foreground objects (e.g., adjust the posture of a dog with the appearance and background maintained). [*8*] further explores the roles played by the Q-K-V in encoding semantic information and observes that Q determines the semantic meaning of each spatial location while K-V offer the context of different parts of the image for weighted aggregation.
>
> Scene Text Editing shares a common motivation with aforementioned non-rigid editing task [*7,8*] but preserves differences in its complicated non-convex structure, wherein a minor stroke discrepancy can significantly alter visual perception and lead to misinterpretation. The complicated text structure brings obstacles to the replacement since it's difficult to either obtain a text mask or mask the text region precisely on the intermediate feature. In addition, without the mask, performing replacement on the whole K-V often results in sparse attention maps during self-attention which may lead to inaccurate transfers and artifacts on final result.
>
> To deal with the problem, we prefer a heuristic integration of K-V from two branches to avoid losing concentration and enable a flexible weighted aggregation for style consistency control. Experience in ***Paper Tab.4*** verifies that our integration strategy surpasses replacement strategy [*7*] on maintaining text style consistency. We believe the integration strategy would also be helpful when performing editing without a precise mask for the complex structural object.
>
> > **References**
>
> [1] Van et al. Neural discrete representation learning. NeurIPS, 2017.
>
> [2] Peng et al. UPOCR: Towards Unified Pixel-Level OCR Interface. ICML, 2024.
>
> [3] Wang et al. What is the Real Need for Scene Text Removal? Exploring the Background Integrity and Erasure Exhaustivity Properties. TIP, 2023.
>
> [4] Peng et al. Viteraser: Harnessing the power of vision transformers for scene text removal with segmim pretraining. AAAI, 2024.
>
> [5] Zhang et al. A Tale of Two Features: Stable Diffusion Complements DINO for Zero-Shot Semantic Correspondence. NeurIPS, 2023.
>
> [6] Tang et al. Emergent Correspondence from Image Diffusion. NeurIPS, 2023.
>
> [7] Cao et al. MasaCtrl: Tuning-Free Mutual Self-Attention Control for Consistent Image Synthesis and Editing. CVPR, 2023.
>
> [8] Alaluf et al. Cross-Image Attention for Zero-Shot Appearance Transfer. ACM SIGGRAPH, 2024.

---

### Official Review · Reviewer_Ktb6 · 2024-07-12

**Soundness:** 3
**Presentation:** 3
**Contribution:** 3
**Rating:** 6
**Confidence:** 4

**Summary:**

This manuscript proposes a diffusion-based scene text editing method with prior guidance control. It incorporates style-structure guidance into the model to enhance the text style consistency and rendering accuracy. A Glyph-adaptive mutual self-attention mechanism is designed to deconstruct the implicit fine-grained features to improve the generation quality. Besides, a new real-world image-pair dataset is proposed for fair comparisons. The experimental results show that the proposed method achieves the best results among existing methods.

**Strengths:**

**Innovative Approach**: The paper introduces a novel diffusion-based method that leverages prior guidance control for text editing, addressing the limitations of previous GAN-based and diffusion-based methods.

**Fine-grained Style Disentanglement**: By constructing a fine-grained text style disentanglement, the method improves text style consistency, which is crucial for maintaining the original style and texture during editing.

**Robust Glyph Structure Representation**: The system incorporates a robust representation of text glyph structures, enhancing the accuracy of text rendering.

**Comprehensive Evaluation**: The authors have created the ScenePair dataset to evaluate both visual quality and rendering accuracy, providing a more holistic assessment of STE methods.

**Weaknesses:**

**Limitation in Task Scope**: The proposed method, while highly specialized in the editing domain, is limited to the task of text editing and does not encompass text generation capabilities. This contrasts with previous methods that offer a dual functionality of both generation and editing. The inability to generate new text content as well as edit existing text may be considered a significant limitation, restricting the method's applicability in scenarios that require creative text synthesis in addition to editing.

**Insufficient Ablation Study**: Although the proposed method decomposes text style processing into four distinct tasks—text color transfer, text font transfer, text removal, and text segmentation—the paper does not present a comprehensive ablation study that systematically evaluates the contribution of each individual task. An ablation study is crucial for understanding the impact of each component on the overall performance. Without it, the discussion lacks depth regarding the significance and interplay of these tasks in achieving the method's objectives. Therefore, further research is needed to dissect the individual contributions and optimize the balance between these tasks for improved performance and efficiency.

**Questions:**

1. Can the proposed method deal with generation task directly, as mentioned in Weaknesses?

2. The ablation study for the four tasks is needed, as mentioned in Weaknesses.

**Limitations:**

Yes

---

> ### Author Rebuttal · Authors · 2024-08-07
>
> Thank you for the detailed review. We are encouraged by the comments that TextCtrl serves as an innovative approach and ScenePair provides a more holistic assessment to STE methods. The concerns are taken care to address point by point in the following.
>
>
> > **[W1/Q1]: Limitation in task scope to offer a dual functionality of both generation and editing.**
>
> [A1]: Thanks for bringing the question and concern. To start off, we would like to highlight that the main focus of our work is to address scene text editing with throughly disentangled text style and glyph structure feature as the prior guidance. Differ from scene text generation, scene text editing possesses the unique challenge in faithfully representing text styles (e.g., font, color, serifs) as well as detail texture (e.g., grids, shadow). To this end, we constructe a style encoder for explict text style extraction and propose the glyph-adaptive mutual self-attention machanism for implicit consistency control, with the concentration on preserving fidelity in editing.
>
> Some inpainting-based methods offer a dual functionality of both generation and editing, yet their editing ability mainly inherits from reasoning the text style according to surrounding unmasked context, which would be uncontrollable and unreliable sometimes and lead to style deviation, as shown in the ***Uploaded Pdf Fig.A***.
>
> Although our current model focuses on high-fidelity text editing, it can also be easily modified to enable both high-fidelity text editing and unconditional creative text generation. Specifically, leveraging the classifier-free guidance (CFG) technique [*1*] which enables a joint training of conditional and unconditional model
>
>  $\widehat\epsilon_{\theta}(t, z_{t}, c) = \omega ·\epsilon_{\theta}(t,z_t, c) + (1-\omega)·\epsilon_{\theta}(t, z_t, \emptyset)$,
>
>  we could encode either the pure background image or the text removal output of style encoder, serving as an additional guidance $c_{bg}$.  Along with the text style guidance $c_{style}$, we have
>
>  $\widehat\epsilon_{\theta}(t, z_{t}, c_{bg}, c_{style}) = \omega ·\epsilon_{\theta}(t,z_t, c_{bg}, c_{style}) + (1-\omega)·\epsilon_{\theta}(t, z_t, c_{bg}, \emptyset)$,
>
>  based on which our model could generate without $c_{style}$, which offers a dual functionality for generation and editing. This multi-condition approach has been proven to be feasible in other tasks [*2,3*]. We appreciate the reviewer's suggestion on model versatility and we will further the research in future work.
>
>
>
>
>
> > **[W2/Q2] Further ablation study is needed to evaluate the contribution of each task in text style pretraining.**
>
> [A2]: We have performed an ablation study of the text style pretraining from a broad perspective in  ***Paper Tab.3*** to verify its joint contribution to foster style representation. Yet we agree that a further discussion on the contribution of each task would benefit the understanding of each component and optimization in implementation.
>
> The text style pretraining is decomposed into four sub-tasks, wherein *Text Font Transfer* and *Text Color Transfer* focus on textural style (e.g., glyph style, text color), while *Text Removal* and *Text Segmentation* concentrate on capturing spatial style (e.g.,  background, size). Due to the time constraint of rebuttal, we augment our style ablations with further experiments on two groups of sub-tasks, namely texture group (font & color) and spatial group (removal & segmentation). Results show that texture group mainly achieves growth in fidelity reflected on FID, while spatial group improves the overall quality of editing reflected on all metrics. The reason lies in the fact that sub-tasks in texture group concentrate on detail texture transfer, while sub-tasks in spatial group reckon with both foreground text and background pixel, thus achieving better statistical similarity. Meanwhile, the two groups jointly contribute to the mutual collaboration in training and achieve finer performance.
>
>
> |Font|Color|Removal|Seg|SSIM $\uparrow$|PSNR $\uparrow$|MSE $\downarrow$|FID  $\downarrow$|
> |:------:|:------:|:------:|:------:|------|------|------|------|
> |&#10008;|&#10008;|&#10008;|&#10008;|0.3130|14.78|0.0475|66.10|
> |&#10004;|&#10004;|&#10008;|&#10008;|0.3197|14.79|0.0470|58.81|
> |&#10008;|&#10008;|&#10004;|&#10004;|0.3652|14.91|0.0453|47.19|
> |&#10004;|&#10004;|&#10004;|&#10004;|0.3756|14.99|0.0447|43.78|
>
>
>
> > **References**
>
> [1] Ho et al. Classifier-Free Diffusion Guidance. NeurIPS Workshop, 2021.
>
> [2] Sharma et al. Alchemist: Parametric Control of Material Properties with Diffusion Models. CVPR, 2024.
>
> [3] Song et al. Arbitrary Motion Style Transfer with Multi-condition Motion Latent Diffusion Model. CVPR, 2024.

---

> > ### Comment · Reviewer_MkTr · 2024-08-13
> > **The rebuttal has been read**
> >
> > After reviewing the feedback and rebuttal, I would like to maintain my rating.

---

> > > ### Author Response · Authors · 2024-08-13
> > >
> > > We sincerely appreciate your time and effort throughout the review. Thank you again for your valuable comments to improve the quality of our manuscript!

---

### Official Review · Reviewer_MkTr · 2024-07-13

**Soundness:** 3
**Presentation:** 3
**Contribution:** 3
**Rating:** 5
**Confidence:** 5

**Summary:**

The paper addresses the challenges of Scene Text Editing (STE) by introducing a diffusion-based STE method, TextCtrl. Traditional GAN-based STE methods struggle with generalization, and existing diffusion-based methods face issues with style deviation. TextCtrl overcomes these limitations through two main components: (1) Style-Structure Guidance: By disentangling fine-grained text styles and robust text glyph structures to improve text style consistency and rendering accuracy. (2) Glyph-adaptive Mutual Self-attention Mechanism: it enhances style consistency and visual quality by reconstructing implicit fine-grained features of the source image.
Additionally, the paper introduces ScenePair, the first real-world image-pair dataset designed for STE evaluation.

**Strengths:**

(1) Style-Structure Guidance: By disentangling fine-grained text styles and robust text glyph structures to improve text style consistency and rendering accuracy.

(2) Glyph-adaptive Mutual Self-attention Mechanism: it enhances style consistency and visual quality by reconstructing implicit fine-grained features of the source image.

(3) This paper propose an image-pair dataset termed ScenePair to enable the comprehensive assessment on real-world images.

**Weaknesses:**

(1) In this paper, the authors mention that ‘their style guidance predominantly originates from the image’s unmasked regions, which can be unreliable in complex scenarios and fail in style consistency.’ In theory, the unmasked region can provide the editing model with more style prior information from the surrounding environment, which is beneficial for style transfer during the editing process.

(2) An important characteristic in text editing is to ensure that the edited word is compatible with the features of the surrounding unmasked region of the original image. Similar approaches have already emerged, such as TextDiffuser and AnyText, as mentioned in your paper. Therefore, the setting of editing only cropped images does not guarantee this compatibility. Additionally, the method proposed in the paper involves relatively complex design choices.

(3) About the ScenePair benchmark: how do authors ensure that the backgrounds of image pairs are consistent? Or are there some examples where the image-pair’s backgrounds are noticeably mismatched after your filtering rules?

(4) About the evaluation: During the evaluation of inpainting-on-full-size-image method DiffSTE, TextDiffuser, and AnyText, do authors replace the unedited region with the origin image when comparing with the cropped-based method.

**Questions:**

Please see weaknesses section.

**Limitations:**

Please see weaknesses section.

---

> ### Author Rebuttal · Authors · 2024-08-06
>
> Thank you for the comprehensive review, which will certainly help improve the revised paper. The concerns are taken care to address point by point in the following.
>
>
> > **[W1]: Discussion of more style prior information from unmasked region.**
>
> [A1]: We agree that surrounding unmasked region could also provide style prior. Nevertheless, how to determine the informative unmasked region for providing accurate style prior is still an unresolved problem.
>
> Current inpainting-based methods [*1,2,3*] take the whole masked image as input and reason the text style by surrounding unmasked region. As shown in ***Uploaded Pdf Fig.A***, this implicit and indirect style learning strategy lack controllability and reliability, leading to text style deviation sometimes. In contrast, TextCtrl focuses on cropped text region which provides direct and explicit guidance for maintaining style consistency in editing.
>
>
>
> > **[W2]: Editing on cropped images does not guarantee compatibility; The method involves relatively complex design choices.**
>
> [A2]:  We would like to clarify that while inpainting-based methods possess the intrinsic ability for smoothening, cropped-based methods adopt a background preservation policy to guarantee compatibility.
>
> Compatibility of TextCtrl is ensured through two aspects. Firstly, a text removal task is involved in style pretraining to explicitly foster the ability to preserve backgrounds. Secondly, during inference, we integrates K-V pairs from reconstruction branch to alleviate background degradation. Quantitative results on SSIM in ***Paper Tab.1*** verify the compatibility of TextCtrl comprehensively, along with visualization in ***Paper Appendix Fig.11***.
>
> Though specialized design has been proposed in TextCtrl, the framework is simple and detachable. It's built upon a vanilla text-to-image diffusion model [*6*] (SD v1-5), with a replacement of text encoder and an increment of style encoder. Our design focuses on scene-text-oriented pretraining and inference control, while following vanilla training and sampling settings in [*6*].  The aforementioned settings jointly contribute to TextCtrl without bringing much complexity.
>
>
>
> > **[W3]: About ScenePair: consistency measurement of backgrounds; mismatched examples after the filtering rules.**
>
> [A3]: The collection pipeline is depicted in ***Paper Appendix Fig.8***. To ensure pair consistency, we design an automatic pairing algorithm with scoring function:
>
> $Score = \lambda_{length} * S_{length} + \lambda_{ratio}*S_{ratio} + \lambda_{distance} * S_{distance} + \lambda_{SSIM} * S_{SSIM}$,
>
> wherein $S_{length}$ is the similarity of text lengths, $S_{ratio}$ is the similarity of aspect ratio, $S_{distance}$ is the centre distance and $S_{SSIM}$ is the structure similarity of cropped-text regions. The consistency of backgrounds is ensured technically by $S_{SSIM}$ and empirically by $S_{distance}$ (nearer texts are more likely to share same background and lighting condition).
>
> After pairing, pairs with $Score$ higher than a threshold are selected and in most cases they share a common background as shown in ***Paper Appendix Fig.9***. Still, we manually filter out a very few number of unsatisfied pairs to guarantee final quality. Examples are provided in ***Uploaded Pdf Fig.B***.
>
>
> > **[W4]: About full-size-image evaluation: do authors replace the unedited region with the origin image for inpainting-based methods?**
>
> [A4]: Official code of inpainting-based methods [*1,2,3*] is adopted and the output is directly used for full-image evaluation without replacing unedited regions, since it is generally considered as the final results which intrinsically distinguishes inpainting-based method.
>
> Yet we acknowledge the necessity of evaluation with the unedited region replaced by origin images for a comprehensive comparison. To this end, we extend the full-size evaluation, wherein *(w)* and *(w/o)* denote with and without replacing unedited regions. Comparing within each inpainting-based method, a large margin emerges indicating the inpainting strategy downgrades the image quality of unmasked region, which mainly result from the lossy compression of VAE and the model capacity. Comparison across different methods verifies the compatibility achieved by TextCtrl. Besides, it is worth noting that cropped-image evaluation is regarded as the main indicator of text style consistency since full-image evaluation spends most calculation on unedited regions.
>
> | Methods          |SSIM $\uparrow(\times10^{-2})$               | FID $\downarrow$ |
> | ---      | ---                                | ---      |
> | SRNet [*4*]               |98.91                              |1.48                  |
> | MOSTEL [*5*]              | 98.96                             |1.49                  |
> | DiffSTE [*1*] *(w/o)*     | 76.91                             |96.78                  |
> | DiffSTE [*1*] *(w)*       | 98.86                             |2.37                  |
> | TextDiffuser [*2*] *(w/o)*| 92.76                             |12.23                  |
> | TextDiffuser [*2*] *(w)*  | 98.97                             |1.65                  |
> | AnyText [*3*] *(w/o)*     | 82.57                             |16.92                  |
> | AnyText [*3*] *(w)*       | 98.99                             |1.93                  |
> | TextCtrl                  | **99.07**                             |**1.17**                  |
>
>
> > **References**
>
> [1] Ji et al. Improving Diffusion Models for Scene Text Editing with Dual Encoders. TMLR, 2024.
>
> [2] Chen et al. TextDiffuser: Diffusion Models as Text Painters. NeurIPS, 2023.
>
> [3] Tuo et al. AnyText: Multilingual Visual Text Generation And Editing. ICLR, 2024.
>
> [4] Wu et al. Editing Text in the Wild. ACM MM, 2019.
>
> [5] Qu et al. Exploring Stroke-Level Modifications for Scene Text Editing. AAAI, 2023.
>
> [6] Rombach et al. High-Resolution Image Synthesis with Latent Diffusion Models. CVPR, 2022.

---

### Author Rebuttal · Authors · 2024-08-06

We thank all reviewers for their thoughtful and constructive feedback. It's encouraged to hear from the reviewers that

- The Model TextCtrl: *"Innovative Approach; addresses weak correlation between text prompts and glyph structures; generate edited image with both high style fidelity and high recognition accuracy;"* [Reviewer Ktb6, mqKG, 25Re]

- The Benchmark ScenePair: *"enable the comprehensive assessment on real-world images; providing a more holistic assessment of STE methods; is contributed to the community;"* [Reviewer MkTr, Ktb6, rTPt, mqKG, 25Re]

- The Evaluation: *"Comprehensive; extensive and well-discussed; demonstrate notable performance improvement;"* [Reviewer Ktb6, rTPt, mqKG]

- The Paper: *"is well written; the presentation is clear;"* [Reviewer rTPt, 25Re]

In response to the review, we provide individual replies below to address the remaining concerns from each reviewer. Notably, some concerns share a common topic yet are raised from different perspectives, to which our respective responses could serve as a mutual reference.

- Research and discussion of text style disentanglement pretraining; [A2 for Reviewer Ktb6, A1 for Reviewer rTPt, A5 for Reviewer 25Re]

- Reason and detail for ControlNet module involved in ablation study; [A1 for Reviewer mqKG, A4 for Reviewer 25Re]


Besides, an additional uploaded one-page pdf, some references and the main paper are used in our responses for clarity and precision.

- Figures from uploaded one-page pdf are denoted as ***Uploaded Pdf Fig.X***;

- Figures and tables from the main paper are denoted as ***Paper Fig.X/Tab.X***;

- Figures from the appendix of the main paper are denoted as ***Paper Appendix Fig.X***;

- Reference paper is denoted as [*X*];

We hope our responses could resolve your concerns. Please do not hesitate to let us know if you have further questions.

---

### Decision · Program_Chairs · 2024-09-25

**Decision:**

Accept (spotlight)

**Comment:**

All five reviewers recommend accepting this paper (2 “Accept”, 1 “Weak Accept” and 2 “Borderline Accept”). The reviewers mentioned that the key idea of the proposed method is novel, the constructed ScenePair dataset is valuable, and the results of this paper are impressive. The AC agrees with the reviewers that the paper is interesting and deserves to be published in NeurIPS 2024.